# Mitigation of PM[2.5] and Ozone Pollution in Delhi: A Sensitivity Study during the Pre-monsoon period

**Ying Chen[1,2*], Oliver Wild[1,2], Edmund Ryan[1,9], Saroj Kumar Sahu[4], Douglas Lowe[5], Scott Archer-Nicholls[6], Yu Wang[5], Gordon McFiggans[5], Tabish Ansari[1], Vikas Singh[7], Ranjeet S. Sokhi[8], Alex Archibald[6], Gufran Beig[3]**

[1]Lancaster Environment Centre, Lancaster University, Lancaster, LA1 4YQ, UK

[2]Data Science Institute, Lancaster University, Lancaster, LA1 4YW, UK

[3]Indian Institute of Tropical Meteorology, Pune, India

[4]Environmental Science, Dept. of Botany, Utkal University, Bhubaneswar, India

[5]Centre for Atmospheric Sciences, School of Earth, Atmospheric and Environmental Sciences, University of Manchester, Manchester, UK

[6]NCAS-Climate, Department of Chemistry, University of Cambridge, Cambridge, UK

[7]National Atmospheric Research Laboratory, Gadanki, AP, India

[8]Centre for Atmospheric and Climate Physics Research, University of Hertfordshire, Hatfield, Hertfordshire, UK

[9]School of Mathematics, University of Manchester, Manchester, UK

*Correspondence to: Ying Chen (y.chen65@lancaster.ac.uk)*

**Abstract:**

Fine particulate matter ($PM_{2.5}$) and surface ozone ($O_3$) are major air pollutants in megacities such as Delhi, but the design of suitable mitigation strategies is challenging. Some strategies for reducing $PM_{2.5}$ may have the notable side-effect of increasing $O_3$. Here, we demonstrate a numerical framework for investigating the impacts of mitigation strategies on both $PM_{2.5}$ and $O_3$ in Delhi. We use Gaussian process emulation to generate a computationally efficient surrogate for a regional air quality model (WRF-Chem). This allows us to perform global sensitivity analysis to identify the major sources of air pollution, and to generate emission-sector based pollutant response surfaces to inform mitigation policy development. Based on more than 100,000 emulation runs during the pre-monsoon period (peak $O_3$ season), our global sensitivity analysis shows that local traffic emissions from Delhi city region and regional transport of pollutions emitted from the National Capital Region surrounding Delhi (NCR) are dominant factors influencing $PM_{2.5}$ and $O_3$ in Delhi. They together govern the $O_3$ peak and $PM_{2.5}$ concentration during daytime. Regional transport contributes about 80% of the $PM_{2.5}$ variation during the night. Reducing traffic emissions in Delhi alone (e.g., by 50%) would reduce $PM_{2.5}$ by 15-20% but lead to a 20-25% increase in $O_3$. However, we show that reducing NCR regional emissions by 25-30% at the same time would further reduce $PM_{2.5}$ by 5-10% in Delhi and avoid the $O_3$ increase. This study provides scientific evidence to support the need for joint coordination of controls on local and regional scales to achieve effective reduction in $PM_{2.5}$ whilst minimizing the risk of $O_3$ increase in Delhi.

## 1. Introduction

Exposure to air pollutants increases morbidity and mortality (Huang et al., 2018a;WHO, 2013). The urban air quality in India, especially in Delhi, is currently among the poorest in the world (WHO, 2013, 2016a, b). In addition to the local impacts, the Indian monsoon can transport air pollutants to remote oceanic regions, inject them into the stratosphere and redistribute them globally (Lelieveld et al., 2018). This makes the impact of Indian air pollution wide ranging regionally and globally and it has interactions with climate and ecosystems world-wide (Menon et al., 2002;Gao et al., 2019).

PM$_{2.5}$ (particulate matter with an aerodynamic diameter of less than 2.5 μm) is a major air pollutant, causing increases in disease (Pope et al., 2009;Gao et al., 2015;Stafoggia et al., 2019) and reduced visibility (Mukherjee and Toohey, 2016;Wang and Chen, 2019;Khare et al., 2018). The population of India experiences high PM$_{2.5}$ exposure, and this is responsible for ~1 million premature deaths per year (Conibear et al., 2018;Gao et al., 2018). Residential emissions are estimated to contribute ~50% of PM$_{2.5}$ concentrations and to cause more than 0.5 million annual mortalities across India (Conibear et al., 2018). Previous studies reported an annual averaged PM$_{2.5}$ loading of 110-140 μg/m$^3$ in Delhi during 2015-2018, leading to ~10,000 premature deaths per year in the city (Chen et al., 2019; Chowdhury and Dey, 2016; WHO, 2016a). In Delhi, the traffic sector (~50%) and the domestic sector (~20%) are the major local contributors to PM$_{2.5}$ (Marrapu et al., 2014). Efforts to control traffic emissions in Delhi in recent years by introducing an alternating 'odd-even' licence plate policy have led to reductions in PM$_{2.5}$ of less than 10% (Chowdhury et al., 2017). This indicates that there is an urgent need for a coordinated plan to mitigate PM$_{2.5}$ pollution (Chowdhury et al., 2017).

Surface ozone (O$_3$), another major air pollutant, is damaging to health and reduces crop yields (Ashworth et al., 2013;Lu et al., 2018;Kumar et al., 2018). The risks of respiratory and

cardiovascular diseases are increased from short-term exposure to high ambient $O_3$ and from long-term exposure at low levels (WHO, 2013;Turner et al., 2016;Fleming et al., 2018). Oxidation of volatile organic compounds (VOCs) in the presence of nitrogen oxides (NOx) is the main source of surface ozone. Rapid economic development in India has greatly increased the emissions of these $O_3$ precursors (Duncan et al., 2016), leading to significant increases in $O_3$ especially during the pre-monsoon period (Ghude et al., 2008). Hourly maximum $O_3$ reaches as much as 140 ppbv during the pre-monsoon season in Delhi (Ghude et al., 2008), comparable to the most polluted regions in China (150 ppbv, Wang et al., 2017) and higher than the most polluted areas in the U.S. (110 ppbv, Lu et al., 2018).

Mitigation of $PM_{2.5}$ pollution may lead to an increase in surface ozone, because the dimming effect of aerosols and removal of hydroperoxy radicals are reduced, facilitating $O_3$ production (Huang et al., 2018b;Li et al., 2018;Hollaway et al., 2019). Furthermore, co-reduction of NOx and $PM_{2.5}$ emissions may increase $O_3$ in cities where $O_3$ production is in a VOC-limited photochemical regime (Ran et al., 2009;Xing et al., 2018;Xing et al., 2017). This has recently been reported in a number of Asian megacities, e.g. Shanghai (Silver et al., 2018), Beijing (Wu et al., 2015;Liu et al., 2017;Chen et al., 2018) and Guangzhou (Liu et al., 2013). Delhi and coastal cities in India, which are known to be VOC-limited (Sharma et al., 2017), may face increased $O_3$ as a side-effect of emission controls focused on $PM_{2.5}$. Therefore, studies of mitigation strategies that target both $PM_{2.5}$ and $O_3$ are urgently needed (Chen et al., 2018), particularly as urban air pollution in India has been much less well studied than in many other countries.

To investigate the impacts of mitigation strategies with respect to both $PM_{2.5}$ and $O_3$, we demonstrate a framework for generating emission-sector based pollutant response surfaces using Gaussian process emulation (O'Hagan and West, 2009;O'Hagan, 2006). The response

surfaces describe that how the pollutants, i.e., $PM_{2.5}$ and $O_3$, will respond to the changes in emissions from different sectors. We conduct global sensitivity analysis to identify the dominant emission sectors controlling $PM_{2.5}$ and $O_3$, and then generate sector based response surfaces to quantify the impacts on $PM_{2.5}$ and $O_3$ of emission reductions. In contrast to simple sensitivity analysis varying one input at a time, this allows full exploration of the entire input space, accounting for the interactions between different inputs (Pisoni et al., 2018;Saltelli et al., 1999). Conventionally, chemical transport models (CTMs) are used to calculate the impacts on pollutants concentrations of different mitigation scenarios. However, the computational expensive of CTMs makes them unsuitable for performing global sensitivity analysis or generating response surfaces, which usually require thousands of model runs. To overcome this difficulty, source-receptor relationships (Amann et al., 2011) or computational efficient surrogate models, trained on a limited number of CTM simulations, are used to replace the expensive CTM. These approaches have been used to perform sensitivity and uncertainty analysis of regional air quality models (Pisoni et al., 2018), assessment of regional air quality plans (Zhao et al., 2017;Xing et al., 2017;Pisoni et al., 2017;Thunis et al., 2016) and sensitivity and uncertainty analysis of global and climate simulations (Ryan et al., 2018;Lee et al., 2016;Lee et al., 2012). Here, we use surrogate model to explore the sensitivity of $PM_{2.5}$ and $O_3$ on sector-based emission controls in Delhi, for developing a mitigation strategy addressing both pollutants.

In this study, we demonstrate the value of such a framework for supporting decision makers in determining better mitigation strategies. We give examples of its use in investigating impacts of mitigation scenarios on $PM_{2.5}$ and $O_3$ pollutions in Delhi, and demonstrate that regional joint coordination of emission controls over National Capital Region (NCR) of Delhi is essential for an effective reduction of $PM_{2.5}$ whilst minimizing the risk of $O_3$ increase.

## 2. Materials and Methods

 **2.1 WRF-Chem Model Baseline Simulation**

WRF-Chem (v3.9.1) – an online, fully coupled chemistry transport model (Grell et al., 2005) – has been widely used in previous studies of air quality across India (Marrapu et al., 2014;Mohan and Gupta, 2018;Gupta and Mohan, 2015;Mohan and Bhati, 2011). The model has also been used to estimate the health burden (Conibear et al., 2018;Ghude et al., 2016) and reduction in crop yields (Ghude et al., 2014) from the exposure to $PM_{2.5}$ and $O_3$ over India.

In this study, we focus on the hot and dry pre-monsoon period in Delhi, when average temperatures are around 32 $^o$C and relative humidity (RH) is about 35% (Ojha et al., 2012). $O_3$ approaches its annual peak in pre-monsoon due to strong solar radiation (Ghude et al., 2008;Ojha et al., 2012). During the pre-monsoon period, desert dust can contribute significantly to particulate matter in Delhi (Kumar et al., 2014b;Kumar et al., 2014a). Here, we perform WRF-Chem simulation for the period of 2–15 May 2015 (with two additional days for spin-up), when Delhi was not significantly influenced by dust storms according to MODIS observations (https://earthdata.nasa.gov/earth-observation-data/near-real-time/hazards-and-disasters/dust-storms). Strong dust storms started to influence the Indo Gangetic Plain on 21-24 April and 19 May 2015, respectively. This minimizes the uncertainties resulting from dust storm simulation and permits a stronger focus on anthropogenic emissions. Resuspended dust from road traffic is also a major contributor to $PM_{2.5}$ in Delhi, and this is estimated and included in the emission inventory as described below.

The model configuration follows the study of Marrapu et al. (2014), and the parameterizations used are listed in Table 1. Three nested domains are used, with coverage of

South Asia (45 km resolution), the Indo Gangetic Plain (15 km resolution), and the National Capital Region (5 km resolution), see Fig. 1. A test simulation with a fourth domain over Delhi at 1.67 km resolution suggests that a further increase in resolution does not substantially improve model performance (details in Text S1), and this is in line with results from a previous study (Mohan and Bhati, 2011). The Carbon Bond Mechanism version Z (CBMZ, Zaveri and Peters, 1999) coupled with the MOSAIC (Zaveri et al., 2008) aerosol module with four size bins is used to represent gaseous chemical reaction and aerosol chemical and dynamical processes. We neglect wet scavenging and cloud chemistry processes here, as the impact of these is likely to be negligible during the dry pre-monsoon period over India. No precipitation was recoded in Delhi during the simulation period.

The initial and boundary conditions for chemical species are provided from MOZART-4 global results (https://www.acom.ucar.edu/wrf-chem/mozart.shtml). Our baseline simulation is driven by European Centre for Medium-Range Weather Forecasts (ECMWF) meteorological data, as we find that this reproduces regional meteorology better than that from the National Centers for Environmental Prediction (NCEP) over India, consistent with a recent study (Chatani and Sharma, 2018). The ECMWF reanalysis dataset (ERA-Interim) assimilates observations with a number of nearly $10^7$ per day (Dee et al., 2011), and is used for grid nudging, initial and boundary conditions for WRF-Chem at horizontal and temporal resolutions of 0.75º × 0.75º and 6 hours, respectively. The wind pattern and temperature over Delhi in May 2015 is generally captured well in simulations driven by either meteorological dataset, but the model captures the variation in relative humidity much better (R=0.7) with ECMWF data than with NCEP data (R=0.4, negative bias of 20-40%). More detailed discussion is provided in Text S2.

The high-resolution Fire Inventory from NCAR (FINN, Wiedinmyer et al., 2011) is adopted to provide biomass burning emissions. Interactive biogenic emissions are included

using the Model of Emissions of Gases and Aerosols from Nature (MEGAN, Guenther et al., 2006). The global Emission Database for Global Atmospheric Research with Task Force on Hemispheric Transport of Air Pollution (EDGAR-HTAP, Janssens-Maenhout et al., 2015) version 2.2 (year 2010) at $0.1° \times 0.1°$ resolution is used to represent anthropogenic emissions apart from over Delhi, where they are represented by a high-resolution monthly inventory for 2015 developed under the System of Air Quality Forecasting and Research (SAFAR) project (Sahu et al., 2011;Sahu et al., 2015). In the absence of a diurnal variation in emissions specific to Delhi, we adopt diurnal variations from Europe in this study (Denier van der Gon et al., 2011). The SAFAR inventory provides emission fluxes of $PM_{10}$, $PM_{2.5}$, black carbon, organic carbon, NOx, CO, $SO_2$ and NMVOC (non-methane volatile organic compounds) from five sectors, including power (POW), industry (IND), domestic or residential (DOM), traffic (TRA) and wind blow dust from roads (WBD). Wind blow dust includes dust resuspended from vehicle movement on paved and unpaved roads (Sahu et al., 2011), and is therefore closely related to traffic emissions, and we combine this into the traffic sector for our study.

The NMVOC emissions are speciated according to the EDGAR (v4.3.2) global inventory (Huang et al., 2017), and are then lumped for the CBMZ chemistry scheme. The speciation mapping is detailed in Table 2 and described below, and a toolkit has been developed to perform this mapping. Emissions of alcohols and ethers are split 20%:80% between methanol and ethanol by mass and then converted to molar emissions with a fractionation based on Murrells et al. (2009). Emissions of paraffin carbon (PAR) are calculated by converting mass emissions from each VOC group to molar emissions and then multiplying by the number of paraffin carbons in order to conserve carbon. Hexanes and higher alkanes are converted to molar emissions of hexane and then multiplied by six to give PAR emissions. Other alkenes are mapped to molar emissions of butane, and this is then apportioned between terminal olefin carbons (OLET), internal olefin carbons (OLEI) and PAR on a molar ratio of 1:1:4 following

(Zaveri and Peters, 1999). Ketones are split 60%:40% by mass between acetone (KET) and methyl-ethyl ketone (MEK), then converted to molar emissions with fractions based on (Murrells et al., 2009). As MEK is not included in the CBMZ mechanism, we apportion molar emissions of MEK equally between KET and PAR.

## 2.2 Observational Network

Air quality and meteorological monitoring networks are operated in Delhi under the SAFAR project coordinated by IITM (Ministry of Earth Sciences, Government of India). Measurements of $PM_{2.5}$, $O_3$ and NOx during the May 2015 simulation period are available from six monitoring stations in Delhi: C V Raman (CVR), Delhi University (DEU), Indira Ghandi International Airport Terminal-3 (AIR), Ayanagar (AYA), NCMRWF (NCM) and Pusa (PUS).

The instruments are calibrated and measurements are quality controlled in the SAFAR project (http://safar.tropmet.res.in); more details are given in previous studies (Sahu et al., 2011;Beig et al., 2013;Aslam et al., 2017). Site locations are shown in Fig. 2 and measured variables are given in Table S1.

## 2.3 Global Sensitivity Analysis of Urban Air Pollution

We perform global sensitivity analysis (GSA) (Iooss and Lemaître, 2015) to quantify the sensitivity of modelled outputs ($PM_{2.5}$ and $O_3$ for this study) to changes in the model inputs, which for this study are emissions from the different emission sectors. One-at-a-time sensitivity analysis is a common way of calculating model sensitivity and involves varying a single model input while the other inputs are fixed at nominal values, e.g., Wild (2007). While one-at-a-time

approach is relatively easy to implement, it assumes that the model response to different inputs is independent and this can lead to biased results (Saltelli et al., 1999;Pisoni et al., 2018;Carslaw et al., 2013). GSA overcomes the problems of the one-at-a-time approach by averaging over the other inputs rather than fixing them at specific values. This allows

calculation of first-order sensitivity indices (SIs) for each variable, corresponding to the $i^{th}$

input variable and the $j^{th}$ output point, is given by the Eq. 1 (Ryan et al., 2018).

$$SI_{i,j} = \frac{Var[E(y_j \mid x_i)]}{Var(y_j)} \times 100\% \qquad (1)$$

where $x_i$ is the $i^{th}$ element of the input; and $y_j$ is the $j^{th}$ element of the output. The 'E(•)' and 'Var(•)' denote the mathematical function that calculate the expectation and variance, respectively. The simplest way of computing $SI_{i,j}$ is by brute force, but this is also the most computationally intensive (Ryan et al., 2018).

The extended Fourier Amplitude Sensitivity Test (eFAST), first developed by Saltelli et al. (1999), is a commonly used approach to perform GSA and calculate SIs and is adopted in this study because of its high efficiency. A basic overview and detailed equations of the eFAST method are given in the section 2.2.2 of Ryan et al. (2018). A challenge to using eFAST is that it typically requires thousands of model runs. To overcome this, we employ a computationally cheaper surrogate model in place of our expensive simulation model WRF-Chem. A surrogate model is a simple model (usually statistical) which can map the inputs to the outputs of the simulation model with sufficiently good accuracy given the same inputs. In this study, we choose a type of surrogate model called a Gaussian process emulator, which works like a function for multi-dimensional interpolation and has been used extensively in many areas of applied science (Carslaw et al., 2013;Koehler and Owen, 1996;Queipo et al., 2005;Vanuytrecht and Willems, 2014;vu et al., 2015;Degroote et al., 2012) and uncertainty assessment of atmospheric models (Lee et al., 2016;Lee et al., 2012;Lee et al., 2011). Gaussian process emulators typically require a relatively small number of runs of the computational-expensive model to generate; this is in contrast to other surrogate modelling approaches, such as neural networks, which typically require thousands of model runs to train them. For a basic overview

of a Gaussian process emulator see O'Hagan (2006), detailed introduction and equations are also given in the section 2.3 of Ryan et al. (2018). Before using the emulator in place of the WRF-Chem model to carry out the thousands of model runs required for GSA, we train the emulator using a relatively small number of WRF-Chem model runs. Following previous studies (Carslaw et al., 2013;Lee et al., 2016), a Maximin Latin hypercube space-filling design is employed to select the designs of training runs for WRF-Chem. Latin hypercube sampling is a statistical method for generating a near-random sample of parameter values from a multidimensional distribution (Shields and Zhang, 2016). Here, we search through 100,000 Latin hypercube random designs to find the optimal one where the parameter space is filled most effectively. This ensures that the sets of inputs chosen cover as large a fraction of the input space as possible. Full details (including R codes) of how to generate the Gaussian process emulator, eFAST method and GSA can be found in Ryan et al. (2018).

In this study, we focus on a limited number of the emission sectors to demonstrate the effectiveness of the approach: domestic/residential emissions in Delhi (DOM), traffic emissions in Delhi (TRA, including WBD), power and industry in Delhi (POW+IND) and total emissions in the National Capital Region outside Delhi (NCR). NCR represents the contribution of regional transport to pollution in Delhi. According to the SAFAR emission inventory, the total $PM_{2.5}$ emissions of DOM, TRA, POW+IND and NCR are about 1.8, 6.1, 3.1 and 8.5 Gg/month in May 2015, respectively. The Gaussian process emulator is trained using 20 executions of the WRF-Chem model, with emission scaling drawn from a variation range of 0-200% for each of the four specified sectors (Table S2). Emulation of the impacts of mitigation scenarios on $PM_{2.5}$ and $O_3$ can be performed in minutes on a laptop, in contrast to simulations with WRF-Chem which require a few days on a high-performance computing cluster. The accuracy of the emulator as a surrogate of WRF-Chem model is evaluated using a 'leave-one-out' cross-validation (Bastos and O'Hagan, 2009). This involves training the

emulator using 19 out of the 20 sets of inputs/outputs from the WRF-Chem model runs and then evaluating the emulator against the 20[th] simulation. This process is carried out for each of the 20 sets of inputs/outputs. Given that the output space is multi-dimensional (i.e. modelled $O_3$ and $PM_{2.5}$ varied spatially and in time), the validation of the emulator by comparing 10,000 (random-samples varied spatially and in time) of emulator output values against the corresponding output values of the WRF-Chem model. The emulator validation plot is shown in Fig. 3. Modelled and emulated $O_3$ and $PM_{2.5}$ lie very close to the 1:1 line with $R^2$ values of more than 95% as shown in Fig. 3, indicating that the emulation provides an accurate representation of the input-output relationship of the WRF-Chem model.

**2.4 Response Surfaces**

Response surfaces are useful for investigating the relationship between model inputs and outputs, in this case between sectoral emissions and modelled pollutant concentrations. They have been widely applied for air quality studies and policy making (EPA, 2006b, a;Zhao et al., 2017;Xing et al., 2017). Here, we analyse the responses of $PM_{2.5}$ and $O_3$ to changes in emissions from each sector of between 0% and 200%. The computationally efficient Gaussian process emulation enables us to generate response surfaces without the computational burden of a large number of runs of the WRF-chem model.

**2.5 Outline of Analysis**

We use the WRF-Chem model to simulate the hourly concentrations of $O_3$ and $PM_{2.5}$ over the Delhi region during 2-15 May 2015 and evaluate the results against observations. We perform a simple sensitivity analysis to investigate the contributions of biomass burning and biogenic emissions to $PM_{2.5}$ and $O_3$ in Delhi. We then conduct a global sensitivity analysis, using the eFAST method (see section 2.3) along with Gaussian process emulation, to determine the sensitivity of modelled $O_3$ and $PM_{2.5}$ concentrations to changes in the dominant

anthropogenic emission sectors. Finally, we generate response surfaces to identify appropriate mitigation strategies for reducing $PM_{2.5}$ while minimizing the risks from $O_3$ increase.

## 3. Results and Discussion

### 3.1 Model Performance

The WRF-Chem model captures the general magnitude and variation in $PM_{2.5}$ well (Fig. 4a), with mean bias and error of about -3.5% and 11%, respectively, and an index of agreement (Willmott et al., 2012) of 75%. The frequency distributions of modelled $PM_{2.5}$ are also similar to the observations, with differences in mean and median concentrations of less than 10%, although high concentration spikes are missed by the model (Fig. S1). The modelled $PM_{2.5}$ peaks around 09:00 local time (LT) because the rush hour enhances traffic emissions before the planetary boundary layer (PBL) height has increased (Fig. 4a). This is also seen in the modelled results at DEU (Fig. S2), which is closer to a motorway and shows a more intense $PM_{2.5}$ peak in the morning rush hour. $PM_{2.5}$ is overestimated during the morning rush hour (around 09:00 am) and underestimated during the early morning (03:00-05:00 LT, Fig. 4a and Fig. S2). This may suggest that there is an earlier rush hour or more traffic activity at night in Delhi than in European cities, since we adopted European diurnal emission patterns in this study in the absence of local information. Detailed studies of traffic emissions and their variation in Delhi would help improve these model simulations.

The modelled chemical composition of $PM_{2.5}$ is shown in Fig. S3. Secondary inorganic aerosol (SIA), including sulphate, nitrate and ammonium, only contributes ~25% of aerosol mass in our simulation. In the absence of particle inorganic composition measurements during the simulation period, we compare our results with a previous modelling study of Delhi during the post-monsoon season (Marrapu et al., 2014), which also shows a ~25% contribution of SIA

to PM$_{2.5}$ loading, in line with our results. Furthermore, our results are also consistent with an

observational study, which reported the mass fraction of organic matter (usually calculated as

1.4 times OC) and elemental carbon (usually equivalent to black carbon in modelling studies,

Chen et al., 2016b) in PM$_{2.5}$ of ~20% and ~6% in Delhi during May 2015, respectively (Sharma

et al., 2018).

The model well captures the peak O$_3$ with a bias of less than 5%, although it

underestimates O$_3$ during night-time (Fig. 4b). In general, the diurnal pattern and magnitude of

O$_3$ are captured by WRF-Chem (Fig. 4b), with normalized mean bias and error of about -20%

and 35%, respectively, and an index of agreement of 65%. The underestimation during night-

time is likely to be because NOx is overestimated by a factor of 2-3 at night (Fig. S4), and the

excess NO depletes O$_3$. This is indicated by the frequency distribution of O$_3$ and NOx (Fig.

S5), where the median values of observed O$_3$ and NOx are matched well by the model.

However, the higher peaks of modelled NOx concentration lower the modelled O$_3$ levels,

indicating that Delhi is in VOC-limited photochemical regime. Similar results are found at

AYA (Fig. S6). The larger underestimation of O$_3$ at NCM (Fig. S5d, industrial environment

site) suggests that NOx emission from the industry sector may be overestimated.

**3.2 Impacts of Biogenic and Biomass Burning Emissions**

Before exploring the importance of the four selected anthropogenic emission sectors on

PM$_{2.5}$ and O$_3$ in Delhi during simulation period, we investigate the contributions from other

factors (biomass burning and biogenic emissions). We turn off these sources in the WRF-Chem

simulation and find that there is a negligible contribution from biogenic emissions to PM$_{2.5}$

concentrations over Delhi in this season (Fig. 4c and 4d). It is worth noting that biogenic

emissions may contribute to secondary organic aerosol (SOA) in Delhi, but the formation of

SOA is not represented well by the CBMZ-MOSAIC chemistry-aerosol mechanisms used in

this study. However, this weakness is not expected to have a major influence on our pre-monsoon results; as described above, the difference of organic matter fraction between simulation and observation (Sharma et al., 2018) in May 2015 is less than 5%. About 10% of $PM_{2.5}$ in Delhi is derived from biomass burning during the simulation period. Crop burning in Haryana and Punjab states is a major source of this (Jethva et al., 2018;Cusworth et al., 2018). In contrast, there is a negligible contribution from biomass burning to $O_3$. However, there is a 15-20% contribution to $O_3$ from biogenic emission of VOCs, highlighting that $O_3$ production in Delhi is strongly VOC-limited.

### 3.3 Effect of the Diurnal Variation in Emissions

In order to investigate the competing influences of meteorology and human activities on the diurnal patterns of $PM_{2.5}$ and $O_3$ over Delhi, we test the effect of removing the diurnal variation in anthropogenic emissions ('noDiurnal', see Fig. 4c and 4d). Modelled $PM_{2.5}$ concentrations are very similar to the 'baseline' run during daytime when the PBL is well developed, with differences of less than 5%. This suggests that meteorological processes such as vertical mixing, advection and transport are the dominant factors controlling $PM_{2.5}$ in the daytime. In contrast, freshly emitted pollutants are trapped at night when the PBL is shallow, and concentrations are very sensitive to the emission flux, so that the diurnal pattern of emissions is the dominant factor at night. The $PM_{2.5}$ concentration is almost doubled in the early morning (03:00-09:00 LT, Fig. 4c) when the PBL is shallow and emissions in the 'noDiurnal' case are higher. There is also a large increase in NOx in the early morning (Fig. S4), which leads to greater depletion of $O_3$ (Fig. 4d). However, the concentration of $O_3$ is about 20-25% higher during the ozone peak hour in the afternoon in the 'noDiurnal' case, as the daytime NOx emissions are less (Fig. S4). This sensitivity test also highlights the VOC-limited nature of $O_3$ production in Delhi.

**3.4 Sensitivity Analysis of Pollutants in Delhi**

The importance of each anthropogenic emission sector to pollutant concentrations in Delhi is investigated using global sensitivity analysis and indicated by global sensitivity indices (SIs), as shown in Fig. 5. The sensitivity index is a measure of the contribution of the variation in pollutants from one emission sector to the total variation across all four sectors considered here. A larger SI indicates a larger influence from the corresponding sector to the modelled average surface $PM_{2.5}$ or $O_3$ over Delhi City Region (marked in Fig. 2) in this study.

The $PM_{2.5}$ concentration is most sensitive to emissions from the NCR region surrounding Delhi, with a sensitivity index higher than 50% most of time (Fig. 5a) and reaching 80-90% and ~60% during 03:00-07:00 LT and 12:00-17:00 LT, respectively. During the rush hours in the morning and evening, the sensitivity to NCR emissions is lower, while the sensitivity to Delhi traffic emissions increases by ~30%. Around 10:00 LT, local traffic emissions and emissions from NCR have a similar effect on $PM_{2.5}$. In contrast, local traffic emissions dominate the $PM_{2.5}$ in Delhi around 20:00 LT, with a sensitivity contribution of up to ~80%. This is caused by the collapse of the PBL in the evening rush hour at around 20:00 LT which enhances the sensitivity to fresh local emissions. Local traffic emissions contribute ~60% of primary $PM_{2.5}$ emission in Delhi (Fig. 6a), which remains concentrated in the PBL during rush hours. In contrast, the fully developed PBL in the daytime mixes air down from the free troposphere (Chen et al., 2016a), where regional transport of pollutants from NCR can be important. This could explain the second peak in the sensitivity to NCR emissions (50-60%) during the afternoon (Fig. 5a).

The variation of $O_3$ in Delhi City Region is overwhelmingly dominated by local traffic emissions with a sensitivity index higher than 80% at night-time (Fig. 5b), when $O_3$ and traffic emissions are anti-correlated. Traffic contributes ~75% of total NOx emission in Delhi (Fig.

6b), and the shallow PBL during the night traps the NOx. This removes $O_3$ through chemical reaction in the absence of solar radiation. As the PBL develops in the morning, the sensitivity of $O_3$ to traffic decreases and the sensitivity to NCR emissions increases. The sensitivity to NCR emissions reaches its highest point (70%) when the PBL is fully developed around 15:00 LT. As discussed above, the downward mixing of air from the free troposphere and dilution of local emissions in the fully developed PBL could be the reason for this. The $O_3$ peak coincides with the highest PBL at this time because photolysis and development of the PBL are both driven by solar radiation. The development of the PBL increases the contribution from regional transport, and precursors emitted from the NCR are one of the dominant contributors to the peak of $O_3$ in Delhi. NOx, mainly originating from traffic emissions, is underestimated by ~30% during the $O_3$ peak period (Fig. S4). This uncertainty can propagate into the Gaussian process emulator and could lead to underestimation of the influence of traffic on peak $O_3$, but is not expected to change the nature of our conclusion about the predominance of regional transport and local traffic emissions. In addition, it is noteworthy that the NOx-rich urban plume from Delhi has a substantial influence on $O_3$ in downwind regions across the NCR as well, as discussed in Text S3.

### 3.5 Mitigation Strategies

To demonstrate a framework for developing better mitigation strategies for addressing both $PM_{2.5}$ and $O_3$ pollution in Delhi, emission-sector based pollutant response surfaces are generated using Gaussian process emulation (Fig. 7). For local emissions in Delhi, we focus mainly on traffic and residential sectors here, because we find that power and industrial emissions have a more limited influence on $PM_{2.5}$ and $O_3$ concentrations in Delhi (Fig. 5). A range of different mitigation strategies are analysed, aiming at mitigating $PM_{2.5}$ pollution whilst minimizing the risk of $O_3$ increase.

We find that the responses of $PM_{2.5}$ and $O_3$ to each emission sector are nearly linear in Delhi. The response surfaces show that reducing local traffic emissions in Delhi leads to an efficient decrease in $PM_{2.5}$ loading (Fig. 7a) but increases $O_3$ greatly (Fig. 7b). Reducing local domestic emissions decreases $PM_{2.5}$ loading less than reducing traffic but without increasing $O_3$. The small impact on $O_3$ may be because domestic emissions are not a major source of NOx, contributing only 15% of that from traffic (Fig. 6). A 10-20% reduction in NOx is expected when reduce local domestic emissions by 50%; but a 35-45% reduction is seen for a 50% reduction in local traffic emissions (Fig. S7). In addition, VOC is reduced more than NOx when controlling domestic emissions, as the VOC/NOx emission ratio (kg/kg) is 1.8 in contrast to a ratio of 0.4 for traffic emissions. Greater reduction of VOC suppresses the increase of $O_3$ in Delhi, which is a VOC-limited environment. A reduction in local traffic emissions alone of 50% could decrease Delhi $PM_{2.5}$ loading by 15-20%, but this would also increase $O_3$ by 20-25%. We note that our model may underestimate the influence of traffic emissions on $O_3$ to some extent as described above (section 3.4), suggesting that the ozone increase could be stronger than we predict. To prevent the side-effect of increasing $O_3$ by controls on traffic emissions, regional cooperation would be required to reduce emissions in the NCR region surrounding Delhi by 25-30%, which also permits a further reduction of $PM_{2.5}$ by 5-10% (Fig. 7c and 7d). This is consistent with a recent study showing that ~60% of $PM_{2.5}$ in Delhi originates from outside (Amann et al., 2017). We test this by performing an additional run with WRF-Chem using emission reductions of 50% and 30% for sectors of local traffic and the surrounding NCR region, respectively. We compare the WRF-Chem results of the additional run and the base case (without change of emissions) against the corresponding results from Gaussian process emulator (Fig. S8). We find that the $PM_{2.5}$ and $O_3$ results from the model runs lie within 5% of those estimated with the emulator and with $R^2$ higher than 95%, demonstrating the high quality of the emulation approach adopted here and underlining its deeper value for identifying

mitigation approaches. The suggested regional joint mitigation with NCR surrounding Delhi is in line with a recent study for mitigating $PM_{2.5}$ in Beijing, which showed that regional coordination over the North China Plain could lead to a reduction in $PM_{2.5}$ of up to 40% in winter (Liu et al., 2016).

## 4. Summary

Previous studies have shown that emission controls focusing on mitigation of $PM_{2.5}$ may result in substantial increases of surface ozone over urban areas that are in VOC-limited photochemical environment. Comprehensive studies of mitigation strategies with respect to both $PM_{2.5}$ and $O_3$ are urgently required, but are limited in India. In this study, we demonstrate a numerical framework for informing emission-sector based mitigation strategies in Delhi that account for multiple pollutants.

By using Gaussian process emulation with an air quality model (WRF-Chem), we generate a computational efficient surrogate model for performing global sensitivity analysis and calculating emission-sector based pollutant response surfaces. These enable us to exhaustively investigate the impacts of different mitigation scenarios on $PM_{2.5}$ and $O_3$ in Delhi, which help decision makers choose better mitigation strategies. Global sensitivity analysis shows that pollutants originating from the National Capital Region (NCR) surrounding Delhi and local traffic emissions are the major contributors of $PM_{2.5}$ and $O_3$ in Delhi. They co-dominate the $O_3$ peak and $PM_{2.5}$ in Delhi during daytime, while the regional transport governs $PM_{2.5}$ during the night, in line with a recent study showing that ~60% of $PM_{2.5}$ in Delhi originates from outside (Amann et al., 2017). Controlling local traffic emissions in Delhi would have the notable side effect of increasing $O_3$, at least in the pre-monsoon/summer period (peak $O_3$ season) that we consider here. This is in line with recent increases in $O_3$ seen in China (Silver et al., 2018;Li et

al., 2018). The Chinese experience suggests that regional joint coordination is required to effectively mitigate PM2.5 pollution in Beijing (Liu et al., 2016). Our pollutant response surfaces go one step further and suggest that joint coordinated emission controls with the NCR region surrounding Delhi would be required to not only achieve a more ambitious reduction of $PM_{2.5}$ but also to minimize the risk of $O_3$ increases. In the regional joint coordination,

residential energy use could be a dominant emission sector over a large region in India (Conibear et al., 2018).

## 5. Discussion

The experiences of developed countries (Dooley, 2002;EPA, 2011) and recently in China

(Huang et al., 2018a;Wang et al., 2019) show that regional joint coordination can be achieved by changing energy infrastructure (e.g., replacing fossil fuel by renewable energy and natural gas), desulphurisation and denitrification technologies, popularization of new energy vehicles, strict control of vehicle exhaust and reducing road and construction dust. Further studies with more detailed information on specific emission sectors and strategies for clean-technology

development and popularization would permit deeper insight into air pollution mitigation approaches suitable for Delhi. These are needed to address both $PM_{2.5}$ which has a higher impact on public health (e.g., Huang et al., 2018a), and $O_3$ which greatly impacts regional ecology and agriculture (e.g., Avnery et al., 2011). A more comprehensive evaluation of the health and economic benefits of different mitigation strategies would greatly help Indian

decision makers, and the framework we have demonstrated here would provide a strong foundation for this.

## Author contributions

OW and YC conceived the study. YC performed the simulations and emulation, and processed and interpreted the results with help from YW. ER designed and built the Gaussian Process emulator. GB and SKS provided the observations and SAFAR emission inventory. DL., AA, SAN. and GM help pre-process the emission data and develop the emission toolkit. VS and RSS provided useful discussion on the emission inventory. RSS led the development of the PROMOTE project. YC and OW wrote the manuscript with inputs from all co-authors.

## Notes

The authors declare no competing financial interest.

## Acknowledgments

This work was supported by the NERC/MOES/Newton Fund supported PROMOTE project (grant number NE/P016405/1 and NE/P016480/1). The work of E. Ryan was supported by the NERC (grant number NE/N003411/1). The Indian Institute of Tropical Meteorology, Pune, is supported by the Ministry of Earth Science, Government of India. The observations and high-resolution emission inventory are provided by the SAFAR project under MoES (http://safar.tropmet.res.in). The authors appreciate the efforts of the entire team involved in PROMOTE and SAFAR projects. The paper is based on interpretation of scientific results and in no way reflect the viewpoint of the funding agency.

## Data availability

NCEP FNL operational model global tropospheric analyses (ds083.2) were downloaded from https://rda.ucar.edu/data/ds083.2/, and sea surface temperature data were downloaded from http://polar.ncep.noaa.gov/sst/. ECMWF interim reanalyses (ERA-Interim) were downloaded from http://apps.ecmwf.int/datasets/data/interim-full-daily. MOZART-4 global model results are downloaded from http://www.acd.ucar.edu/wrf-chem/mozart.shtml. FINN biomass burning emissions dataset is downloaded from http://bai.acom.ucar.edu/Data/fire/. Toolkits for emission processing are available from https://github.com/douglowe/WRF_UoM_EMIT/releases/tag/v1.0 and https://github.com/douglowe/PROMOTE-emissions/releases/tag/v1.0 .

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

**Table 1.** Configuration of WRF-Chem

| Physics | WRF option |
|---|---|
| Micro physics | Lin scheme (Lin, 1983) |
| Surface Layer | MM5 similarity |
| Boundary layer | YSU (Hong, 2006) |
| Cumulus | Grell 3D |
| Urban | 3-category UCM |
| Shortwave radiation | Goddard shortwave (Chou, 1998) |
| Longwave radiation | Rapid Radiative Transfer Model |
| **Chemistry and Aerosol** | **Chem option** |
| Gas-phase mechanism | CBMZ (Zaveri and Peters, 1999) |
| Aerosol module | MOSAIC with 4 bins (~40 nm to 10 µm) (Zaveri et al., 2008) |
| Photolysis rate | Fast-J photolysis scheme (Wild et al., 2000) |
| **Emissions Inventories** | |
| Anthropogenic Emissions | SAFAR-2015 Delhi and EDGAR-HTAP v2.2 |
| Biogenic Emissions | MEGAN (Guenther et al., 2006) |
| Biomass Burning Emissions | FINN (Wiedinmyer et al., 2011) |

**Table 2.** Map of NMVOC from EDGAR emission to CBMZ scheme.

| EDGAR Name | Description | CBMZ [mol] |
| --- | --- | --- |
| VOC1 | Alcohols | 20% $CH_3OH$ |
| | | 80% $C_2H_5OH$ |
| VOC2 | Ethane | $C_2H_6$ |
| VOC3 | Propane | PAR*3 |
| VOC4 | Butane | PAR*4 |
| VOC5 | Pentane | PAR*5 |
| VOC6 | Hexanes + other Alkanes | PAR*6 |
| VOC7 | Ethene | ETH |
| VOC8 | Propene | OLET+PAR |
| VOC9 | Ethyne | PAR*2 |
| VOC10 | Isoprene | ISOP |
| VOC11 | Monoterpenes | ISOP*2 |
| VOC12 | Other Alkenes | OLEI*0.5+OLET*0.5+PAR*2 |
| VOC13 | Benzene | TOL-PAR |
| VOC14 | Toluene | TOL |
| VOC15 | Xylenes | XYL |
| VOC16 | Trimethylbenzenes | XYL+PAR |
| VOC17 | Other Aromatics | XYL+PAR |
| VOC18 | Esters | RCOOH |
| VOC19 | Ethers | 20% $CH_3OH$ |
| | | 80% $C_2H_5OH$ |
| VOC21 | Formaldehyde | HCHO |
| VOC22 | Other Aldehydes | ALD2 |
| VOC23 | Ketones | 60% KET |
| | | 40% KET+PAR |
| VOC24 | Alkanoic Acids | RCOOH |

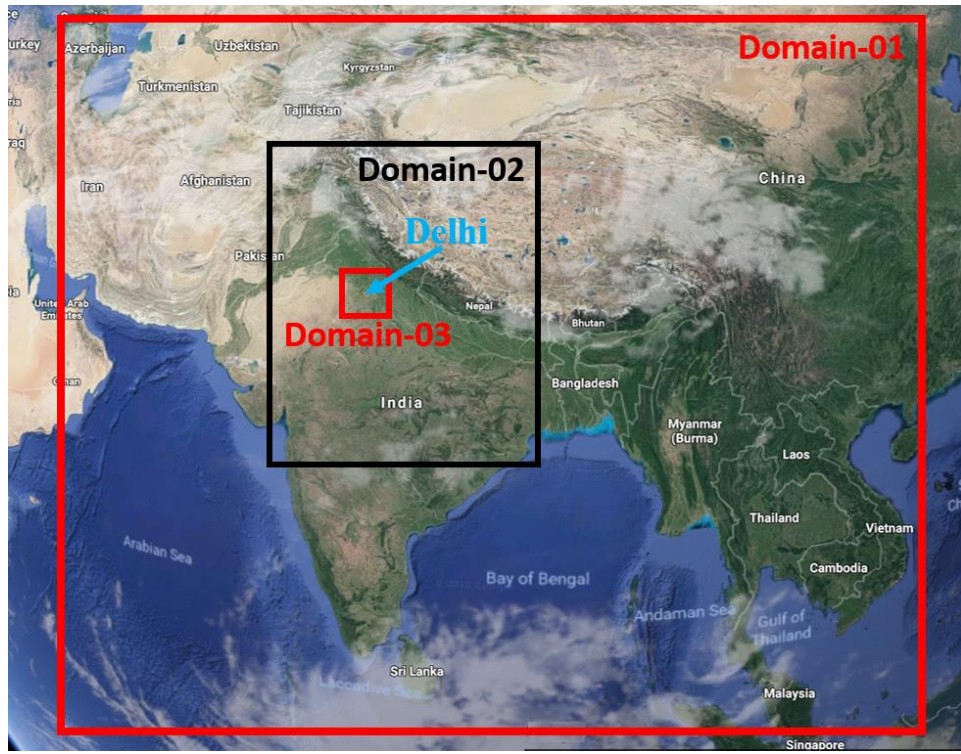

**Figure 1.** Map of simulation domains, modified from Google Earth.

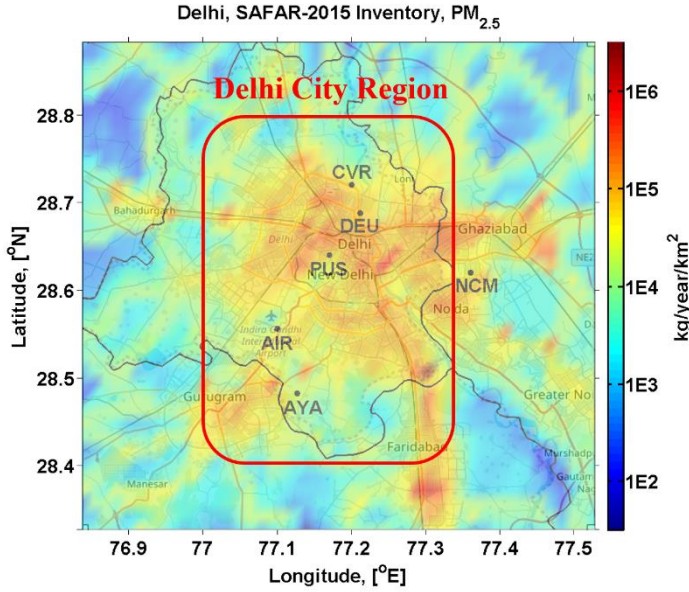

**Figure 2.** SAFAR inventory of total PM$_{2.5}$ emission. The locations of measurement sites over Delhi are marked by black dots, and the Delhi City Region is marked by a red box.

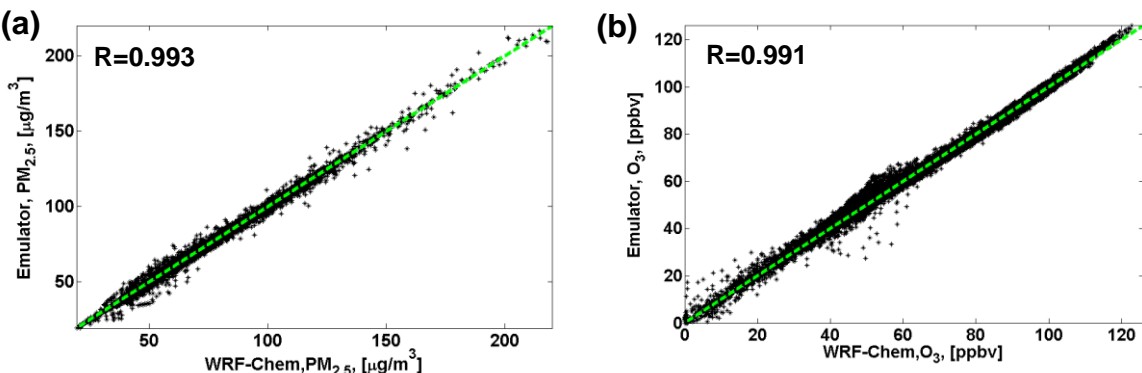

**Figure 3.** Validation of Gaussian process emulator with WRF-Chem model. (**a**) PM$_{2.5}$; (**b**) O$_3$. The green dashed line indicates the 1:1 line.

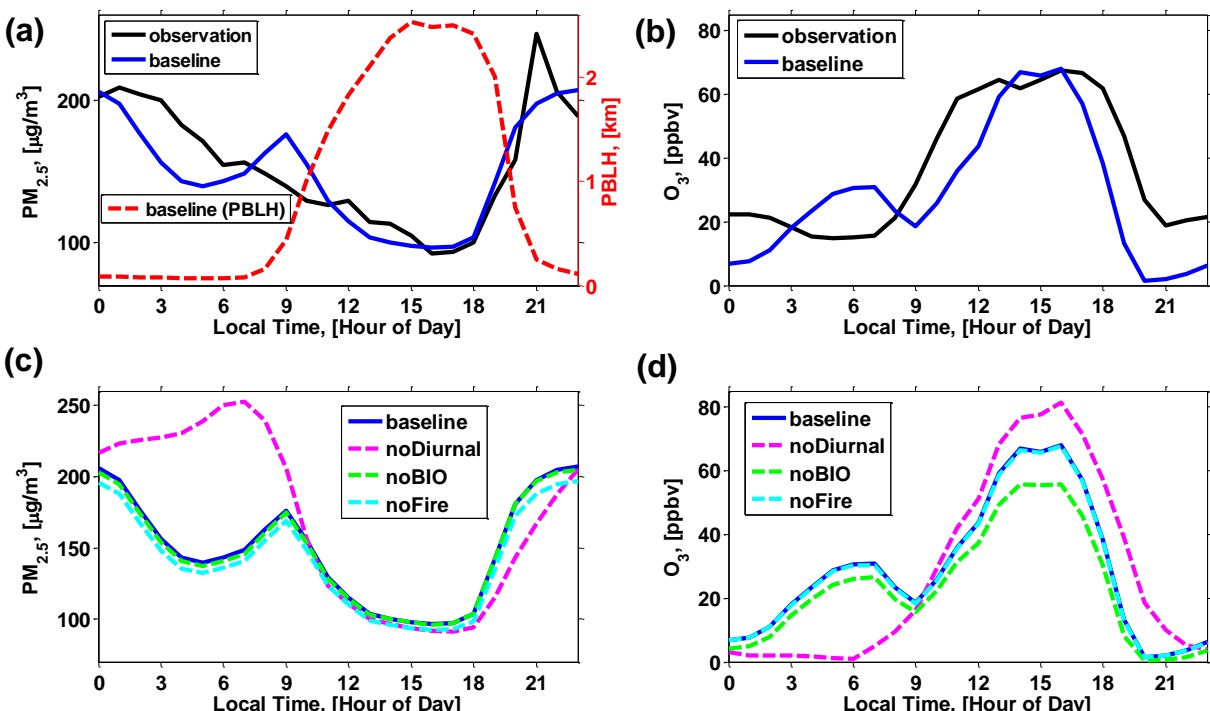

**Figure 4.** Average diurnal patterns of pollutants during the 2-15 May 2015 simulation period. (**a**) Modelled and observed PM$_{2.5}$ and model PBL height (PBLH); (**b**) O$_3$; (**c**) results of sensitivity studies for PM$_{2.5}$; (**d**) results of sensitivity studies for O$_3$. The left panels (a, c) are for site CVR, and the right panels (b, d) are for site AIR (marked in Fig. 2). The sensitivity runs 'noFire' and 'noBIO' show model results without biomass burning and biogenic emissions, respectively; and 'noDiurnal' show model results with constant anthropogenic emissions rates throughout the day.

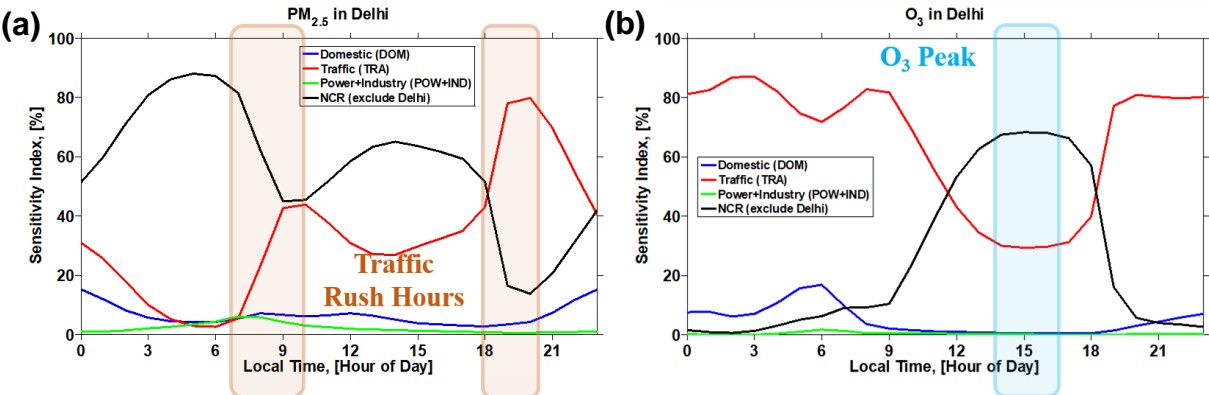

**Figure 5.** Averaged diurnal pattern of global sensitivity indices during the 2-15 May simulation period. (**a**) PM$_{2.5}$; (**b**) O$_3$. The PM$_{2.5}$ and O$_3$ results are averaged over Delhi City Region (marked with red box in Fig. 2). The morning and evening rush hours and the period of peak ozone are marked with the boxes to highlight the notable changes in contribution from each emission sector.

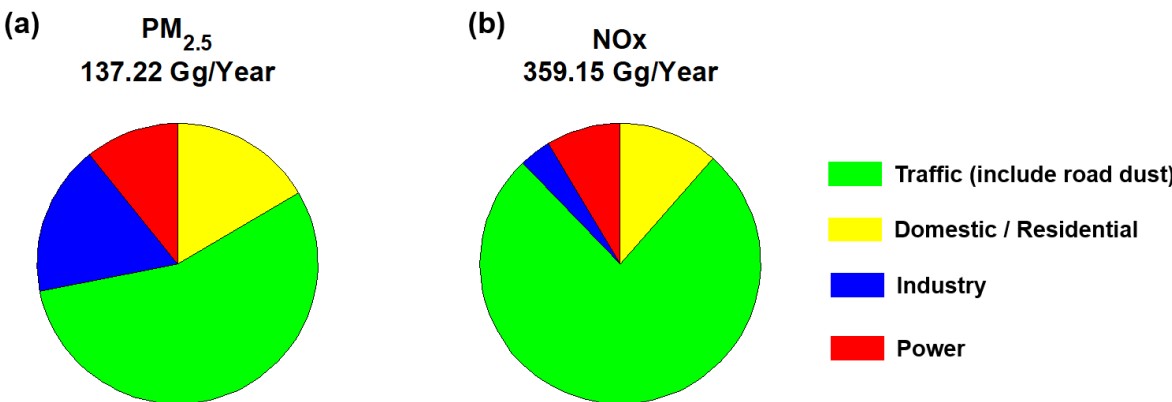

**Figure 6.** Annual emission of different sectors in Delhi from SAFAR inventory. (a) PM2.5; (b) NOx. The emissions of black carbon, organic carbon, non-methane VOC (NMVOC) and SO$_2$ are given in the supplementary Fig. S13~~2~~.

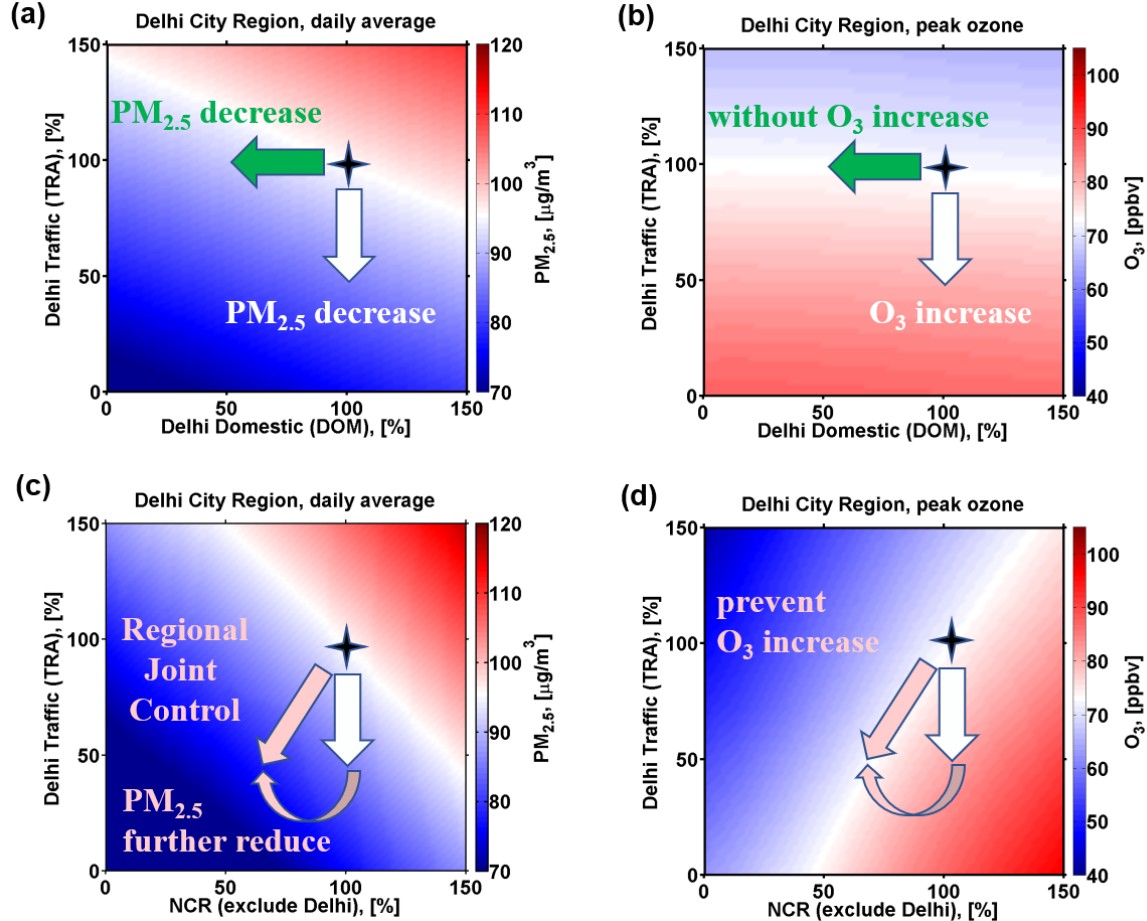

**Figure 7.** Response surfaces for PM$_{2.5}$ and ozone concentrations over Delhi City Region, averaged over 2-15 May 2018. (**a**) Daily average of PM$_{2.5}$ concentrations as a function of local traffic and domestic emissions in Delhi; (**b**) peak hourly ozone concentrations as a function of local traffic and domestic emissions in Delhi; (**c**) daily average of PM$_{2.5}$ concentrations as a function of local traffic emissions in Delhi and emissions in NCR region surrounding Delhi; and (**d**) peak hourly ozone concentrations as a function of local traffic emissions in Delhi and emissions in NCR region surrounding Delhi. The star indicates current conditions and the arrows show the effect of possible emission controls.

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
