# Peer review of "Mitigation of PM2.5 and Ozone Pollution in Delhi: A Sensitivity Study during the Pre-monsoon period"

_Atmospheric Chemistry and Physics, 2019_

## Referee Comment (RC1) · Anonymous Referee #1 · 24 Sep 2019

General Comments

Chen et al. describe the use of a numerical framework for emulation and sensitivity analysis of a regional air quality model in the development of air quality mitigation strategies for the megacity Delhi. They find that a combination of reduction in traffic emissions within the city, combined with simultaneous reductions in all emission sources in the surrounding region would lead to a reduction of PM2.5, while avoiding an increase in ozone. The reduction of traffic emissions from Delhi alone would increase peak ozone in Delhi due to the high emissions of NOx from traffic, with the resultant reduction in ozone due to changes in the O3-NOx titration effect.

These results are certainly plausible, and consistent with previous work. The potential for ozone to increase when high local NOx emissions are decreased has been well understood for decades, as has the transboundary nature of ozone and the corresponding need to control precursor emissions over large spatial scales in order to achieve reductions in ozone. The authors themselves also cite previous work showing that a large fraction of the PM2.5 in Delhi originates outside of the city. I would generally regard the results presented by Chen et al. as unremarkable, and not of sufficient scientific novelty to warrant publication in Atmospheric Chemistry and Physics.

The most novel aspect of the study as I see it is the use of a statistical model emulator, combined with a technique called "global sensitivity analysis" to rapidly discover and evaluate effective emission mitigation options with a minimal amount of computational expense. The paper would potentially have merit if it had more of a technical focus on the methodology. Unfortunately, the methods are not described or evaluated well enough in the present version of the manuscript for me to be able to recommend publication. In order to be recommendable for publication, the manuscript needs major revisions focusing on better description and evaluation of the methods for model emulation and sensitivity analysis. I give suggestions for improving the manuscript in my specific comments, below.

Specific Comments

The introduction is concise and well written, but since the novelty of the paper is in its methodological advances, it needs an expanded discussion of model emulation and global sensitivity analysis.

Line 134: WRF-Chem is an online model, which is capable of calculating its own meteorology. Please describe how the model is "driven" by the ECMWF meteorological data. Are they used as boundary conditions? Is some kind of nudging or data assimilation used?

Line 181: The reference given here (Iooss and Lemaitre, 2015) appears to use "global

sensitivity analysis" as an umbrella term to describe a range of techniques. The authors should be more specific about what kind of global sensitivity analysis they describe in this manuscript.

Line 185: The paper by Saltelli et al. (1999) is behind a paywall. Simply giving a reference to this study is not enough to describe the method they employ. The authors must also give a summary of how this works and how it is specifically employed in their study.

Line 188: Similarly, "Gaussian process emulation" is not sufficiently well described in the manuscript. A summary of how this technique works and how it is applied must be included.

Lines 209-210: "10,000 random samples" are performed to check that the emulator "can fully represent the results of WRF-Chem". Does this mean that the authors performed 10,000 runs of WRF-Chem, and compared them with 10,000 runs of the emulator? Or did they do something else? What do each of the points in Fig. 3 actually represent? This is not clear at all. The authors seem to be relying on this analysis to show that the emulation "provides a good representation of the model", but in my opinion this has not been shown at all. Much more detail is needed here.

Line 256: NOx appears to be significantly underestimated by WRF-Chem during the middle of the day, when peak ozone concentrations are also modelled. Given the central role of NOx as an ozone precursor, it appears that the modelled peak ozone is being well simulated for the wrong reasons. There is likely a compensating error in some other aspect of the model. The authors should provide some discussion about how these errors in WRF-Chem would propagate into their emulator and affect the global sensitivity analysis.

Line 267: "We remove these sources". From what? WRF-Chem itself, or the emulator?

Lines 311-312 and Fig. 5b: It should be pointed out somewhere in the discussion that

the overwhelming dominance of traffic NOx emissions on ozone in Delhi is actually through anti-correlation. Presenting the sensitivity indices of "TRA" and "NCR" together on the same plot is potentially quite misleading unless the authors make it clear that their respective influences have opposite sign.

Section 3.5: This is a nice example of the potential power and utility of the methodology. Figure 7d is an especially clear illustration of the emissions control trajectory which is required to prevent an increase in ozone in Delhi despite reducing local NOx emissions. As mentioned in my general comments, this general approach to emission control (reducing ozone by focusing on regional-scale emissions) is consistent with current understanding of ozone chemistry, so this result by itself is rather unremarkable. What is really interesting here is the ability to rapidly discover an optimal emission mitigation pathway, and quantify its effects. What is missing here though, is a verification that the same combined emission controls for TRA and NCR would result in the same reductions in PM and ozone when employed in the full WRF-Chem model. It would only take one WRF-Chem run to verify this result. In my opinion this extra run is necessary for the authors to be able to show that their approach really is capable of what they claim.

---

## Referee Comment (RC2) · Anonymous Referee #2 · 30 Sep 2019

This manuscript uses Gaussian process emulation to generate a efficient surrogate for WRF-Chem to perform the sensitivity analysis of PM2.5 and O3 to sources, and to provide air pollution mitigation suggestions. The combination of WRF-Chem with Gaussian process emulation is novel to reduce computational complexity for sensitivity analysis. The results, especially the joint control suggestions for PM2.5 and O3, are useful in terms of air pollution control. The manuscript is well written, but some parts of it are not clear enough. I would recommend for publication after the authors address the following specific comments:

Line 45: please add references for this statement: Menon, S., Hansen, J., Nazarenko,

[Figure]

L. and Luo, Y., 2002. Climate effects of black carbon aerosols in China and India. Science, 297(5590), pp.2250-2253. Gao, M., Sherman, P., Song, S., Yu, Y., Wu, Z. and McElroy, M.B., 2019. Seasonal prediction of Indian wintertime aerosol pollution using the ocean memory effect. Science advances, 5(7), p.eaav4157.

Line 81: Is it possible to provide a clear definition of pollutant response surface?

Line 130-132: This statement is a bit general. Better to use measurements of precipitation and clouds to show this point.

Fig. 6: Better to provide similar plots for other important species, such as organics, SO2, etc.

Sect 4: It would be better to compare the results with other similar studies and explain the similarities and differences.

---

## Author Comment (AC1) · 6 Nov 2019

**Response to comments of referee #1**

**General comments**

Chen et al. describe the use of a numerical framework for emulation and sensitivity analysis of a regional air quality model in the development of air quality mitigation strategies for the megacity Delhi. They find that a combination of reduction in traffic emissions within the city, combined with simultaneous reductions in all emission sources in the surrounding region would lead to a reduction of PM2.5, while avoiding an increase in ozone. The reduction of traffic emissions from Delhi alone would increase peak ozone in Delhi due to the high emissions of NOx from traffic, with the resultant reduction in ozone due to changes in the O3-NOx titration effect.

These results are certainly plausible, and consistent with previous work. The potential for ozone to increase when high local NOx emissions are decreased has been well understood for decades, as has the transboundary nature of ozone and the corresponding need to control precursor emissions over large spatial scales in order to achieve reductions in ozone. The authors themselves also cite previous work showing that a large fraction of the PM2.5 in Delhi originates outside of the city. I would generally regard the results presented by Chen et al. as unremarkable, and not of sufficient scientific novelty to warrant publication in Atmospheric Chemistry and Physics. The most novel aspect of the study as I see it is the use of a statistical model emulator, combined with a technique called "global sensitivity analysis" to rapidly discover and evaluate effective emission mitigation options with a minimal amount of computational expense. The paper would potentially have merit if it had more of a technical focus on the methodology. Unfortunately, the methods are not described or evaluated well enough in the present version of the manuscript for me to be able to recommend publication. In order to be recommendable for publication, the manuscript needs major revisions focusing on better description and evaluation of the methods for model emulation and sensitivity analysis. I give suggestions for improving the manuscript in my specific comments, below.

*Many thanks to the reviewer for the comments and suggestions.*

*The chemical relationship between $O_3$ and NOx has been well understood for decades, however, the reduction of $O_3$ pollution is still a troublesome issue for mitigation strategy. For example, recently in China, increase of $O_3$ is attracting increasing concern from the Chinese government*

*and public, despite considerable achievements in controlling PM$_{2.5}$ pollution, as described in the introduction. Only a handful of studies foresee the potential of O$_3$ increase in Delhi under the current mitigation strategy focusing on PM$_{2.5}$ and provide solutions for it. This work provides a quantified map for mitigating PM$_{2.5}$ pollution and tackling O$_3$ increase for Delhi, to avoid the O$_3$ side-effect that China is now facing. Our results could greatly benefit air pollution mitigation with respect to both PM$_{2.5}$ and O$_3$ in Delhi. In addition, as the reviewer mentioned, this work demonstrates a combined approach with WRF-Chem and statistical methods to rapidly discover and evaluate effective emission mitigation options.*

*We agree with the reviewer that more details regarding the methodology can improve this work. We have therefore improved the manuscript accordingly. Please find point-by-point responses below.*

**Specific comments:**

The introduction is concise and well written, but since the novelty of the paper is in its methodological advances, it needs an expanded discussion of model emulation and global sensitivity analysis.

*Thanks for the comments and suggestions. We have improved the manuscript by adding a clearer introduction to these approaches.*

1) Line 134: WRF-Chem is an online model, which is capable of calculating its own meteorology. Please describe how the model is "driven" by the ECMWF meteorological data. Are they used as boundary conditions? Is some kind of nudging or data assimilation used?

*We have added more description about how the model is driven by ECMWF meteorological data, as shown below.*

*"The ECMWF reanalysis dataset (ERA-Interim) assimilates observations with a number of nearly $10^7$ per day (Dee et al., 2011), and is used for grid nudging, initial and boundary conditions for WRF-Chem with horizontal and temporal resolutions of $0.75^o \times 0.75^o$ and 6 hours, respectively."*

2) Line 181: The reference given here (Iooss and Lemaitre, 2015) appears to use "global sensitivity analysis" as an umbrella term to describe a range of techniques. The authors should be more specific about what kind of global sensitivity analysis they describe in this manuscript.

*The reviewer is right that there are many different ways to perform global sensitivity analysis, such as brute force, Sobol method, Fourier Amplitude Sensitivity Test (FAST), random-based-design FAST and extended FAST (eFAST). The sections 1.1 and 2.2 of a recent open-accessible work (Ryan et al., 2018) introduce and summarize well the application and theories/equations of different methods for global sensitivity analysis. In this study, we use the eFAST method to perform global sensitivity analysis. The eFAST method, first developed by Saltelli et al. (1999), is more efficient than the other method mentioned above and has been widely used in diverse areas of science (Carslaw et al., 2013;Koehler and Owen, 1996;Queipo et al., 2005;Vanuytrecht and Willems, 2014;vu et al., 2015).*

*We have rewritten the section 2.3 of our manuscript, added the description and equation for calculating global sensitivity indices, and provided more details about eFAST method for perform the sensitivity indices calculation. Detailed modification of section 2.3 will be shown in the point-4, combining with the responses to the points 2-4.*

3) Line 185: The paper by Saltelli et al. (1999) is behind a paywall. Simply giving a reference to this study is not enough to describe the method they employ. The authors must also give a summary of how this works and how it is specifically employed in their study.

*In this study, we use the eFAST method to perform global sensitivity analysis (GSA). The eFAST method, first developed by Saltelli et al. (1999), is more efficient than the other methods mentioned above and has been widely used in diverse areas of science, as stated above.*

*A detailed introduction to the theory and equations of the eFAST method is given in an open-accessible methodological study (Ryan et al., 2018). We have extensively modified section 2.3 of our manuscript to provide a simple introduction to the approaches adopted, and refer to Ryan et al. (2018) and other previous studies for further details.*

*Description of global sensitivity analysis in the section 2.2 of Ryan et al. (2018):*

*"A common way of conducting global sensitivity analysis for each point in the output space of the simulator – where the output consists of, for example, a spatial map or a time series – is to compute the first-order sensitivity indices (SIs) using variance-based decomposition; this apportions the variance in simulator output (a scalar) to different sources of variation in the different model inputs. Assuming the input variables are independent of one another – which they are for this study – the first-order SI, corresponding to the $i^{th}$ input variable (i = 1, 2, ..., p) and the $j^{th}$ point in the output space, is given by the equation (R1).*

$$SI_{i,j} = \frac{\mathrm{Var}[\mathrm{E}(Y_j \mid X_i)]}{\mathrm{Var}(Y_j)} \times 100 \qquad (R1)$$

*where $X_i$ is the $i^{th}$ column of the n×p matrix X (i.e. a matrix with n rows and p columns) which stores the n samples of p-dimensional inputs, and $Y_j$ is the $j^{th}$ column of the n×m matrix which stores the corresponding n sets of m-dimensional outputs. We multiply by 100 so that the SI is given as a percentage. The notation given by Var(•) and E(•) denotes the mathematical operations that compute the variance and expectation. The simplest way of computing $SI_{i,j}$ is by brute force, but this is also the most computationally intensive."*

*Source from: the section 2.2 of Ryan et al., (2018)*

*Description of eFAST method in the section 2.2.2 of Ryan et al. (2018):*

*"The eFAST method is an alternative and more efficient way of estimating the terms in Eq. (R1). A multi-dimensional Fourier transformation of the simulator f, allows a variance-based decomposition that samples the input space along a curve defined by the equation (R2).*

$$x_i(s) = G_i(\sin(\omega_i s)) \qquad\qquad (R2)$$

*where x =(x1, ..., xp) refers to a general point in the input space that has been sampled, $s \in R$ is a variable over the range (-∞, ∞), $G_i$ is the $i^{th}$ transformation function, and $\omega_i$ is the $i^{th}$ user-specified frequency corresponding to each input. Varying s allows a multi-dimensional exploration of the input space due to the $x_i(s)$ being simultaneously varied. Depending on the simulator, we typically require n=1000–10,000 samples from the input space. After applying the simulator f, the resulting scalar output – denoted generally by y – produces different periodic functions based on different $\omega_i$. If the output y is sensitive to changes in the $i^{th}$ input factor, the periodic function of y corresponding to frequency $\omega_i$ will have a high amplitude."*

**Source from: the section 2.2.2 of Ryan et al., (2018)**

*Please refer to the section 2.2.2 of Ryan et al. (2018) for more specific details about the theory and equations of eFAST method. In order not to replicate many mathematic equations in Ryan et al. (2018), in this study, we have rephrased the section 2.3 to summarize the advantage of global sensitivity analysis compared with the widely used 'one-at-a-time' sensitivity analysis and to simply describe how we perform GSA with non-mathematic language. The detailed modifications of section 2.3 are shown in the point-4, combining with the responses to the points 2-4.*

4) Line 188: Similarly, "Gaussian process emulation" is not sufficiently well described in the manuscript. A summary of how this technique works and how it is applied must be included.

*To perform global sensitivity analysis and generate response surfaces, which describe relationships between the inputs and outputs of models, usually requires thousands of model runs. This is not feasible for a computationally expensive model like WRF-Chem. Therefore, in our study a Gaussian process (GP) emulator, trained by a few model runs, is used as a surrogate of WRF-Chem model. Mathematically, an emulator is a statistical model that mimics*

*the input-output relationship of a simulator, i.e., the expensive WRF-Chem model in this study. The most common form of an emulator is a GP emulator since it has attractive mathematical properties that allow an analytical derivation of the mean and variance of the emulated output (Ryan et al., 2018). As summarized in Ryan et al. (2018):*

*"More formally, a GP is an extension of the multivariate Gaussian distribution to infinitely many variables (Rasmussen and Williams, 2006). The multivariate Gaussian distribution is specified by a mean vector and covariance matrix. A GP has a mean function which is typically given by $m(x) = E(f(x))$ and covariance function given by $c(x, x') = cov(f(x), f(x'))$".*

**Source from: Ryan et al., (2018)**

*Where x and x' are two different input-matrix.*

*Detailed theory of GP emulation is introduced in an open-accessible methodological study (Ryan et al., 2018), and equations and the R code for generating GP emulator are also given in Ryan et al. (2018). In our manuscript, we have substantially revised the section 2.3 to summarize the GP emulator with non-mathematical language, and to describe its advantages and how we use it. The revised section 2.3 is shown below, please also find detailed changes in the revise-tracked file.*

**Revised section 2.3 of this study:**

*"*2.3 Global Sensitivity Analysis of Urban Air Pollution

We perform global sensitivity analysis (GSA) (Iooss and Lemaître, 2015) to quantify the sensitivity of modelled outputs ($PM_{2.5}$ and $O_3$ for this study) to changes in the model inputs, which for this study are emissions from the different emission sectors. One-at-a-time sensitivity analysis is a common way of calculating model sensitivity and involves varying a single model

input while the other inputs are fixed at nominal values, e.g., Wild (2007). While one-at-a-time approach is relatively easy to implement, it assumes that the model response to different inputs is independent and this can lead to biased results (Saltelli et al., 1999;Pisoni et al., 2018;Carslaw et al., 2013). GSA overcomes the problems of the one-at-a-time approach by averaging over the other inputs rather than fixing them at specific values. This allows calculation of first-order sensitivity indices (SIs) for each variable, corresponding to the $i^{th}$ input variable and the $j^{th}$ output point, is given by the Eq. 1 (Ryan et al., 2018).

$$SI_{i,j} = \frac{Var[E(y_j \mid x_i)]}{Var(y_j)} \times 100\%$$ (1)

where $x_i$ is the $i^{th}$ element of the input; and $y_j$ is the $j^{th}$ element of the output. The 'E($\bullet$)' and 'Var($\bullet$)' denote the mathematical function that calculate the expectation and variance, respectively. The simplest way of computing $SI_{i,j}$ is by brute force, but this is also the most computationally intensive (Ryan et al., 2018).

The extended Fourier Amplitude Sensitivity Test (eFAST), first developed by Saltelli et al. (1999), is a commonly used approach to perform GSA and calculate SIs and is adopted in this study because of its high efficiency. A basic overview and detailed equations of the eFAST method are given in the section 2.2.2 of Ryan et al. (2018). A challenge to using eFAST is that it typically requires thousands of model runs. To overcome this, we employ a computationally cheaper surrogate model in place of our expensive simulation model WRF-Chem. A surrogate model is a simple model (usually statistical) which can map the inputs to the outputs of the simulation model with sufficiently good accuracy given the same inputs. In this study, we choose a type of surrogate model called a Gaussian process emulator, which works like a function for multi-dimensional interpolation and has been used extensively in many areas of applied science (Carslaw et al., 2013;Koehler and Owen, 1996;Queipo et al., 2005;Vanuytrecht

and Willems, 2014;vu et al., 2015;Degroote et al., 2012) and uncertainty assessment of atmospheric models (Lee et al., 2016;Lee et al., 2012;Lee et al., 2011). Gaussian process emulators typically require a relatively small number of runs of the computational-expensive model to generate; this is in contrast to other surrogate modelling approaches, such as neural networks, which typically require thousands of model runs to train them. For a basic overview of a Gaussian process emulator see O'Hagan (2006), detailed introduction and equations are also given in the section 2.3 of Ryan et al. (2018). Before using the emulator in place of the WRF-Chem model to carry out the thousands of model runs required for GSA, we train the emulator using a relatively small number of WRF-Chem model runs. Following previous studies (Carslaw et al., 2013;Lee et al., 2016), a Maximin Latin hypercube space-filling design is employed to select the designs of training runs for WRF-Chem. Latin hypercube sampling is a statistical method for generating a near-random sample of parameter values from a multidimensional distribution (Shields and Zhang, 2016). Here, we search through 100,000 Latin hypercube random designs to find the optimal one where the parameter space is filled most effectively. This ensures that the sets of inputs chosen cover as large a fraction of the input space as possible. Full details (including R codes) of how to generate the Gaussian process emulator, eFAST method and GSA can be found in Ryan et al. (2018)."

5) Lines 209-210: "10,000 random samples" are performed to check that the emulator "can fully represent the results of WRF-Chem". Does this mean that the authors performed 10,000 runs of WRF-Chem, and compared them with 10,000 runs of the emulator? Or did they do something else? What do each of the points in Fig. 3 actually represent? This is not clear at all. The authors seem to be relying on this analysis to show that the emulation "provides a good representation of the model", but in my opinion this has not been shown at all. Much more detail is needed here.

*The "10,000 random samples" refers to selection of 10,000 samples from the spatial grid cells and temporal duration of the model run and rebuilding emulators at these points using all but*

*one of the WRF-Chem model training runs, and then comparing these against results from the model run that was omitted in 'leave-one-out' cross-validation. This provides an independent check of how well Gaussian process emulator can represent the results of WRF-Chem. And, thanks to reviewer's suggestion in the last comment, the further validation with the base and regional joint reduction cases also demonstrate good agreement between the emulator and WRF-Chem model (see response to the last comment). We have modified the corresponding context in the last part of section 2.3 to make this clearer, as shown below.*

"We perform 'leave-one-out' cross-validation (O'Hagan and West, 2009;Wang et al., 2011) with 10,000 random samples to check that the Gaussian process emulator can fully represent the results of WRF-Chem."

***Changed to:***

" The accuracy of the emulator as a surrogate of WRF-Chem model is evaluated using a 'leave-one-out' cross-validation (Bastos and O'Hagan, 2009). This involves training the emulator using 19 out of the 20 sets of inputs/outputs from the WRF-Chem model runs and then evaluating the emulator against the 20[th] simulation. This process is carried out for each of the 20 sets of inputs/outputs. Given that the output space is multi-dimensional (i.e. modelled $O_3$ and $PM_{2.5}$ varied spatially and in time), the validation of the emulator by comparing 10,000 (random-samples varied spatially and in time) of emulator output values against the corresponding output values of the WRF-Chem model. The emulator validation plot is shown in Fig. 3. Modelled and emulated $O_3$ and $PM_{2.5}$ lie very close to the 1:1 line with $R^2$ values of more than 95% as shown in Fig. 3, indicating that the emulation provides an accurate representation of the input-output relationship of the WRF-Chem model."

6) Line 256: NOx appears to be significantly underestimated by WRF-Chem during the middle of the day, when peak ozone concentrations are also modelled. Given the central role of NOx as an ozone precursor, it appears that the modelled peak ozone is being well simulated for the wrong reasons. There is likely a compensating error in some other aspect of the model. The

authors should provide some discussion about how these errors in WRF-Chem would propagate into their emulator and affect the global sensitivity analysis.

*Thank you for pointing this out. The choice of y-axis range in Fig. S4 was not fully appropriate and makes NOx looks significantly underestimated by WRF-Chem during the middle of the day when peak ozone occurs. We have changed the y-axis range to [0 200], more in line with convention, as shown below. Instead of a "remarkable underestimation", the NOx is actually underestimated by only ~30%. The daytime variation of NOx (Fig. S4) is directly opposite in pattern to that of boundary layer height (red dashed line in Fig. 4a) between 9 am and 6 pm, suggesting that this underestimation in NOx is closely associated with the variation in boundary layer behavior, a substantial uncertainty in the simulation for which no observational verification is available.*

*The reviewer is right that the good simulation of peak O₃ when NOx is underestimated highlights an uncertainty within the model. This uncertainty would propagate into the emulator and affect the global sensitivity analysis, because emulator learns from the WRF-Chem model and reproduces whatever the model outputs. We have added discussion about this influence in the sections 3.4 and 3.5, as shown below.*

**In the section 3.4 of revised version:**

"NOx, mainly originating from traffic emissions, is underestimated by ~30% during the O₃ peak period (Fig. S4). This uncertainty can propagate into the Gaussian process emulator and could lead to underestimation of the influence of traffic on peak O₃, but is not expected to change the nature of our conclusions about the predominance of regional transport and local traffic emissions."

**In the section 3.5 of revised version:**

"We note that our model may underestimate the influence of traffic emissions on O₃ to some extent as described above (section 3.4), suggesting that the ozone increase could be stronger than we predict. To prevent the side-effect of increasing O₃ by controls on traffic emissions…"

[Figure]

*Original Figure S4.* *Diurnal patterns of NOx concentration from WRF-Chem model and observational results at AIR site. The results are averaged during 02-15 May 2015. Note that 'ECMWF' indicates the model results driven by ECMWF reanalysis data.*

[Figure]

*Revised Figure S4.* *Diurnal patterns of NOx concentration from WRF-Chem model and observational results at AIR site. The results are averaged during 02-15 May 2015. Note that 'ECMWF' indicates the model results driven by ECMWF reanalysis data.*

7) Line 267: "We remove these sources". From what? WRF-Chem itself, or the emulator?

*We have revised the statement, as shown below.*

*"We turn off these sources in the WRF-Chem simulation".*

8) Lines 311-312 and Fig. 5b: It should be pointed out somewhere in the discussion that the overwhelming dominance of traffic NOx emissions on ozone in Delhi is actually through anti-correlation. Presenting the sensitivity indices of "TRA" and "NCR" together on the same plot is potentially quite misleading unless the authors make it clear that their respective influences have opposite sign.

*The reviewer is right that "overwhelming dominance of traffic NOx emissions on ozone in Delhi is actually through anti-correlation.". Although we have already explained this anti-correlation relationship in section 3.4:*

"Traffic contributes ~75% of total NOx emission in Delhi (Fig. 6b), and the shallow PBL during the night traps the NOx. This removes $O_3$ through chemical reaction".

*This may not be clear enough. We have added description at the beginning of this section to make this point clearer, as shown below. Thanks for pointing this out.*

"The variation of $O_3$ in Delhi City Region is overwhelmingly dominated by local traffic emissions with a sensitivity index higher than 80% at night-time (Fig. 5b), where $O_3$ and traffic emissions are anti-correlated."

9) Section 3.5: This is a nice example of the potential power and utility of the methodology. Figure 7d is an especially clear illustration of the emissions control trajectory which is required to prevent an increase in ozone in Delhi despite reducing local NOx emissions. As mentioned in my general comments, this general approach to emission control (reducing ozone by focusing on regional-scale emissions) is consistent with current understanding of ozone chemistry, so this result by itself is rather unremarkable. What is really interesting here is the ability to rapidly discover an optimal emission mitigation pathway, and quantify its effects. What is missing here though, is a verification that the same combined emission controls for TRA and NCR would result in the same reductions in PM and ozone when employed in the full WRF-Chem model. It would only take one WRF-Chem run to verify this result. In my opinion this extra run is necessary for the authors to be able to show that their approach really is capable of what they claim.

*This is a good point. This one extra WRF-Chem simulation makes the verification of emulator more solid and evident. We have performed the extra simulation as suggested. In addition, we also compare the base case (without change of emissions) WRF-Chem results against the emulator results. Noting that the base case is not part of the training cases for emulator, see a list of training cases in Table S2; therefore, this comparison also provides an independent validation for the GP emulator. The results of WRF-Chem and emulator show almost the same reductions in the joint control case. As shown in the figure below, the emulator does reproduce*

*the WRF-Chem simulation nicely, with an uncertainty (normalized mean error) of less than 5%*
*and R² higher than 0.95 for both PM₂.₅ and O₃. Thanks for the suggestion, we have added this*
*extra validation in the section 3.5, as shown below.*

*"We test this by performing an additional run with WRF-Chem using emission reductions of*
*50% and 30% for sectors of local traffic and the surrounding NCR region, respectively. We*
*compare the WRF-Chem results of the additional run and the base case (without change of*
*emissions) against the corresponding results from Gaussian process emulator (Fig. S8). We*
*find that the PM₂.₅ and O₃ results from the model runs lie within 5% of those estimated with*
*the emulator and with R² higher than 95%, demonstrating the high quality of the emulation*
*approach adopted here and underlining its deeper value for identifying mitigation approaches"*

[Figure]

***Newly added Figure S8.*** *Extra validation of Gaussian process emulator results in the*
*mitigation strategy according to Fig. 7. The accuracy of the emulator for reproducing*

*current conditions of PM$_{2.5}$ (**a**) and O$_3$ (**b**), i.e. base case without changing emissions. The accuracy of the emulator for reproducing regional joint coordination conditions of PM$_{2.5}$ (**c**) and O$_3$ (**d**), i.e. NCR joint control case with local traffic emissions reduced by 50% and regional emissions reduced by 30%. All the results are averaged over Delhi City Region, with hourly resolution during the simulation period.*

---

## Author Comment (AC2) · 6 Nov 2019

**Response to comments of referee #2**

**General comments**

This manuscript uses Gaussian process emulation to generate a efficient surrogate for WRF-Chem to perform the sensitivity analysis of PM2.5 and O3 to sources, and to provide air pollution mitigation suggestions. The combination of WRF-Chem with Gaussian process emulation is novel to reduce computational complexity for sensitivity analysis. The results, especially the joint control suggestions for PM2.5 and O3, are useful in terms of air pollution control. The manuscript is well written, but some parts of it are not clear enough. I would recommend for publication after the authors address the following specific comments:

*Many thanks to the reviewer for the comments and suggestions. We have improved the manuscript accordingly. Please find a point-by-point response below.*

**Specific comments:**

1) Line 45: please add references for this statement: Menon, S., Hansen, J., Nazarenko, L. and Luo, Y., 2002. Climate effects of black carbon aerosols in China and India. Science, 297(5590), pp.2250-2253. Gao, M., Sherman, P., Song, S., Yu, Y., Wu, Z. and McElroy, M.B., 2019. Seasonal prediction of Indian wintertime aerosol pollution using the ocean memory effect. Science advances, 5(7), p.eaav4157.

*Thanks for the useful references. We have added these references (Menon et al., 2002;Gao et al., 2019) as suggested.*

2) Line 81: Is it possible to provide a clear definition of pollutant response surface?

*We have added a description of pollutant response surface in the introduction, as shown below.*

*"The response surfaces describe that how the pollutants, i.e., PM$_{2.5}$ and O$_3$, will respond to the changes in emissions from different sectors."*

3) Line 130-132: This statement is a bit general. Better to use measurements of precipitation and clouds to show this point.

*There is no precipitation in Delhi during the simulation period. We have added this statement in the revised version. The climatic averaged monthly precipitation during May in Delhi is only about 20 mm (https://weather-and-climate.com/average-monthly-precipitation-Rainfall,New-Delhi,India).*

4) Fig. 6: Better to provide similar plots for other important species, such as organics, SO2, etc.

*We have added similar plots for black carbon, non-methane VOC, organic carbon and SO₂ in Figure S13, as shown below.*

[Figure]

***Figure S13.*** *Annual emission of different sectors in Delhi from SAFAR inventory. (a) black carbon; (b) organic carbon; (c) non-methane VOC and (d) SO₂.*

5) Sect 4: It would be better to compare the results with other similar studies and explain the similarities and differences.

*Thanks for the suggestion. We have revised the corresponding context in Section 4 to include a comparison with other similar studies. The revised version is shown below, please also see the change-tracked file for more details.*

"They co-dominate the $O_3$ peak and $PM_{2.5}$ in Delhi during daytime, while the regional transport governs $PM_{2.5}$ during the night, in line with a recent study showing that ~60% of $PM_{2.5}$ in Delhi originates from outside (Amann et al., 2017). Controlling local traffic emissions in Delhi would have the notable side effect of increasing $O_3$, at least in the pre-monsoon/summer period (peak $O_3$ season) that we consider here. This is in line with recent increases in $O_3$ seen in China (Silver et al., 2018;Li et al., 2018). The Chinese experience suggests that regional joint coordination is required to effectively mitigate PM2.5 pollution in Beijing (Liu et al., 2016). Our pollutant response surfaces go one step further and suggest that joint coordinated emission controls with the NCR region surrounding Delhi would be required to not only achieve a more ambitious reduction of $PM_{2.5}$ but also to minimize the risk of $O_3$ increases."

**References:**

Amann, M., Purohit, P., Bhanarkar, A. D., Bertok, I., Borken-Kleefeld, J., Cofala, J., Heyes, C., Kiesewetter, G., Klimont, Z., Liu, J., Majumdar, D., Nguyen, B., Rafaj, P., Rao, P. S., Sander, R., Schöpp, W., Srivastava, A., and Vardhan, B. H.: Managing future air quality in megacities: A case study for Delhi, Atmospheric Environment, 161, 99-111, https://doi.org/10.1016/j.atmosenv.2017.04.041, 2017.

Gao, M., Sherman, P., Song, S., Yu, Y., Wu, Z., and McElroy, M. B.: Seasonal prediction of Indian wintertime aerosol pollution using the ocean memory effect, Science Advances, 5, eaav4157, 10.1126/sciadv.aav4157, 2019.

Li, K., Jacob, D. J., Liao, H., Shen, L., Zhang, Q., and Bates, K. H.: Anthropogenic drivers of 2013–2017 trends in summer surface ozone in China, Proceedings of the National Academy of Sciences, 201812168, 10.1073/pnas.1812168116, 2018.

Liu, J., Mauzerall, D. L., Chen, Q., Zhang, Q., Song, Y., Peng, W., Klimont, Z., Qiu, X., Zhang, S., Hu, M., Lin, W., Smith, K. R., and Zhu, T.: Air pollutant emissions from Chinese households: A major and underappreciated ambient pollution source, Proceedings of the National Academy of Sciences, 113, 7756-7761, 10.1073/pnas.1604537113, 2016.

Menon, S., Hansen, J., Nazarenko, L., and Luo, Y.: Climate Effects of Black Carbon Aerosols in China and India, Science, 297, 2250-2253, 10.1126/science.1075159, 2002.

Silver, B., Reddington, C. L., Arnold, S. R., and Spracklen, D. V.: Substantial changes in air pollution across China during 2015–2017, Environmental Research Letters, 13, 114012, 10.1088/1748-9326/aae718, 2018.